# Vitamin D levels and its influencing factors in pregnant women in mainland China: A systematic review and meta-analysis

Bo Chen[1☯], Pengyun Ji[2☯], Qing Wang[3], Wenyan Qin[1☯], Zisheng Li[1]*

1 Department of Nuclear Medicine, Chaohu Hospital Affiliated to Anhui Medical University, Hefei, Anhui, China, 2 Department of Pain, Eastern Theater General Hospital, Xuanwu District, Nanjing, Jiangsu, China, 3 Department of Clinical Laboratory, The Second Affiliated Hospital of Anhui Medical University, Hefei, Anhui, China

☯ These authors contributed equally to this work.
* 919670396@qq.com

## Abstract

### Objective

Maternal vitamin D deficiency is a prevalent public health issue worldwide. While isolated reports from certain cities in China have highlighted the existence of maternal vitamin D deficiency, no nationwide investigation has been conducted on this topic. Therefore, we conducted a meta-analysis and systematic review to examine the prevalence and associated influencing factors of maternal vitamin D deficiency in mainland China. This study aims to provide a theoretical foundation for future prevention and supplementation strategies for maternal vitamin D.

### Methods

We retrieved relevant Chinese and English literature on the status of maternal vitamin D deficiency in mainland China from databases such as CNKI, Wanfang Data, VIP, CBM, Web of Science, Google Scholar, and PubMed. The literature search and database construction were conducted until September 8, 2023. Data were extracted and synthesized following PRISMA guidelines.After literature screening and quality assessment, we performed meta-analysis, sensitivity analysis, and identified publication bias using RevMan 5.3 software.

### Results

A total of 26 articles were reviewed, involving 128,820 pregnant women. Among them, 108,768 had vitamin D insufficiency or deficiency, resulting in a prevalence of 84% (95% CI: 81%~88%). Subgroup analysis revealed the highest prevalence of vitamin D deficiency or insufficiency among pregnant women in mainland China to be in the northwest region (94%, 95% CI: 94%~95%). Furthermore, the highest prevalence was observed during the winter and spring seasons (80%, 95% CI: 77%~83%) and in the early stages of pregnancy (93%, 95% CI: 90%~95%). Significant statistical differences (P<0.05) were found among these

Data Availability Statement: The anonymized data set relevant to this study is publicly available from the EMBL-EBI database at DOI 10.6019/S-BSST1349.

**Funding:** The author(s) received no specific funding for this work.

**Competing interests:** The authors have declared that no competing interests exist.

three subgroups. No publication bias was detected, and sensitivity analysis indicated the stability of the meta-analysis outcome.

## Conclusion

This study provides evidence of the prevalence of vitamin D deficiency or insufficiency among pregnant women in mainland China. To improve the overall health and well-being of the population, relevant health authorities should develop policies aimed at alleviating this phenomenon.

## 1. Introduction

Vitamin D is one of essential nutrients critical to metabolic processes in human body, including the regulation of calcium-phosphorus metabolism, the promotion of bone remodeling and growth, the maintenance of normal cell, tissue and organ development, as well as the prevention of diseases such as rhachitis and osteochondrosis [1]. The active form of vitamin D in the body is known as 1,25-dihydroxyvitamin D. In the human body, vitamin D3 and D2 bind to vitamin D binding protein in the blood plasma and are transported to the liver, where they undergo 25-hydroxylation to become 25-hydroxyvitamin D [25(OH)D] [2].We can obtain vitamin D from dietary sources, however, it is relatively scarce compared to other nutrients, and only a few foods like eggs, fatty fish, beef as well as cod liver oil are rich in vitamin D [3]. The skin is primary vitamin D source, where a precursor of vitamin D3 is converted to vitamin D3 through exposure to UV radiation. Vitamin D3 is then released into the circulation via capillaries [4].

During pregnancy, pregnant women have an increased need for vitamin D. On one hand, this is to meet the mothers' demands and maintain adequate reserves; on the other hand, it is crucial for ensuring proper fetal development. Therefore, pregnancy is the most likely stage of life for vitamin D deficiency [5,6] The concentration of 25-(OH)D remained relatively stable in the whole process of pregnancy [7]. As to the fetus, the mother is the only and sole vitamin D source. During this period, vitamin D status of the mother and the fetus is closely related, and vitamin D supply of the mother has a significant impact on the development and health outcomes of offspring in utero and beyond [8]. As a matter of fact, observational researches from around the world have shown vitamin D is essential for maternal and infant health, and vitamin D deficiency or insufficiency during pregnancy dramatically affects maternal pregnancy and health outcomes [9–12].Vitamin D regulation of the intrauterine immune environment is related to adverse perinatal outcomes, including recurrent pregnancy loss, urinary and reproductive infections, preeclampsia, gestational diabetes, and other pregnancy complications [6]. There have also been reports of possible negative effects on the fetus, such as fetal growth restriction, increased preterm birth risk as well as potential risk of food allergies, asthma, rhachitis, and endocrine and metabolic disorders later in life [13–15].

With the understanding and deepening of various physiological functions of vitamin D in the process of pregnancy, more and more studies have begun to focus on the specific causes of vitamin D deficiency or insufficiency and provide specific strategies for improving vitamin D levels in pregnant women. Various factors, such as reduced dietary intake, decreased endogenous synthesis, inadequate sunlight exposure, as well as illnesses and factors that affect vitamin D absorption or metabolism, might result in the occurrence of vitamin D deficiency or insufficiency [13]. Similarly, maternal knowledge of vitamin D nutrition and increasing vitamin D

sources through multiple pathways can also actively improve vitamin D levels [16]. On the basis of current investigations of vitamin D status in varied populations of pregnant women, almost all researches have indicated generally high prevalence of vitamin D deficiency or insufficiency [10–12]. The meta-analysis together with systematic review summarized the research on the global maternal and infant vitamin D status. Existing data from five regions worldwide indicate that a majority of maternal and infants in almost all areas suffer from vitamin D deficiency [17].

Currently, many regions in China have reported the levels of maternal vitamin D as well as the prevalence of vitamin D insufficiency, deficiency, and adequacy [18–20]. Several researches have also surveyed various factors affecting maternal vitamin D levels [21,22]. However, there is currently no comprehensive nationwide study on this topic. Therefore, the meta-analysis and systematic review were conducted to offer a full introduction to vitamin D status among pregnant women from mainland China. Here we utilized meta-analysis to increase the sample size and integrated relevant data from multiple eligible publications based on different characteristics groups. This approach aims to improve the accuracy of research results and fully analysed vitamin D status among pregnant women in mainland China. The ultimate goal is to provide effective evidence for relevant health authorities in terms of prevention and control measures.

## 2. Materials and methods

We registered our study with PROSPERO (CRD42022335221).Its research basis was the Preferred Reporting Items for Systematic Reviews and Meta-Analyses (PRISMA).

### 2.1 Literature retrieval

A literature search was carried out on PubMed, Embase, Cochrane Library, Google Scholar as well as Web of Science for English-language articles published up to September 8th, 2023. The search terms included "Pregnant Woman," "Woman, Pregnant," "Women, Pregnant," "Pregnant Women," and "Vitamin D," with "China," "Chinese," and "mainland" as population limiters. The search was conducted using subject terms, keywords, and abstracts. In addition, we also searched Chinese authoritative databases such as CNKI, Wanfang and VIP.We based our definitions on the "Guidelines for the Application of Vitamin D in Adults' Bone Health" developed by the Vitamin D Discipline Group of the Osteoporosis Committee of the Chinese Geriatrics Society in 2014. According to these guidelines, a serum 25(OH)D level below 30 nmol/L is considered as vitamin D deficiency, and a level between 30–50 nmol/L is considered as vitamin D insufficiency,and vitamin D levels above 50 nmol/L are considered sufficient.

### 2.2 Literature screening

Inclusion criteria: (1) Researches about serum vitamin D levels in healthy pregnant women; (2) Original articles using cross-sectional or longitudinal cohort study methods; (3) Studies that include the analysis of factors influencing maternal vitamin D levels; (4) Study population consisting of residents in mainland China; (5) Clear research duration, total number of participants (sample size), and criteria for determining vitamin D levels.

Exclusion criteria: (1) Indexes, reviews, conference records, editorials; for duplicate publications, the article with the most complete data relevant to the study was selected; (2) Articles with a quality score lower than 4 points; (3) Full text unavailable; (4) Partial data information in the article is incomplete, suspicious, or contradictory; (5) For studies conducted in the same region and year, the higher-quality article was selected; (6) Studies with a small sample size; (7) Case-control studies.

## 2.3 Literature quality evaluation

The cross-sectional survey study used the quality evaluation tool recommended by the American Agency for Healthcare Research and Quality (AHRQ) [23], which includes 11 evaluation indicators. The options for each indicator are "yes", "no" and "unclear", among which "Yes" is scored as 1 point, "No" and "Don't know" are both scored as 0 points when total score is 11 points. 8 to 11 denote high quality literature, 4 to 7 denote medium quality literature, and 0 to 3 denote low quality literature.

## 2.4 Data extraction

Two researchers independently screened the literature, extracted and cross-checked information. Disagreements, if any, were resolved via consultation or discussion with third parties. As to screening literature, we firstly read text title, then eliminated literature that was obviously irrelevant to our study and further read the abstract and full text to get inclusion. To get data not identified but significant for our study, we contacted original study authors via email or phone if necessary. The following data was extracted from eligible studies: 1) First author, quality assessment score, study field, province/city, and study year; 2) Number of cases with vitamin D insufficiency or deficiency and total survey population; 3) Season of blood collection and gestational week.

## 2.5 Statistical analysis

Statistical analysis was performed via ReviewManager 5.3 software. Incidence served as effect analysis statistic with 95% CI. The χ2 test (test level of α = 0.1) was utilized to analyze heterogeneity among the included studies outcome and combining $I^2$ was employed for quantitative determination of heterogeneity magnitude. If the results of each study showed no statistical heterogeneity, fixed-effects model was employed for Meta-analysis; if the results of each study showed statistical heterogeneity, further analysis of heterogeneity source was performed, and a random-effects model was employed for Meta-analysis, following excluding significant clinical heterogeneity influence. Meta-analysis level was set at α = 0.05. Meta-analyses were performed according to the different regions, periods, seasons, and week-of-gestation dimensions. Subgroup analysis. Funnel plots were used to assess the literature's publication bias. Differences were considered statistically significant at P<0.05.

## 3. Results

### 3.1 Literature search results

Initially, a comprehensive search yielded a total of 633 articles. After a thorough screening process, 26 relevant studies were determined to meet the inclusion criteria for this meta-analysis [2,18–22,24–43]. **Fig 1** showed the selection process. Among 26 references included in our study, 3 studies were assigned a score of 8, 15 for a score of 7, and 8 for a score of 6. Further details regarding the included study characteristics can be found in **Table 1**.

### 3.2 The overall incidence of vitamin D levels in Chinese pregnant

The meta-analysis included 26 studies with 128,820 participants, among which 108,768 were cases of insufficient or deficient vitamin D during pregnancy. The heterogeneity test results demonstrated statistical significance (P<0.1) and moderate heterogeneity ($I^2$>50%), indicating the use of a random effects model for the combination of study outcomes. The meta-analysis revealed overall incidence of insufficient or deficient vitamin D in the process of pregnancy

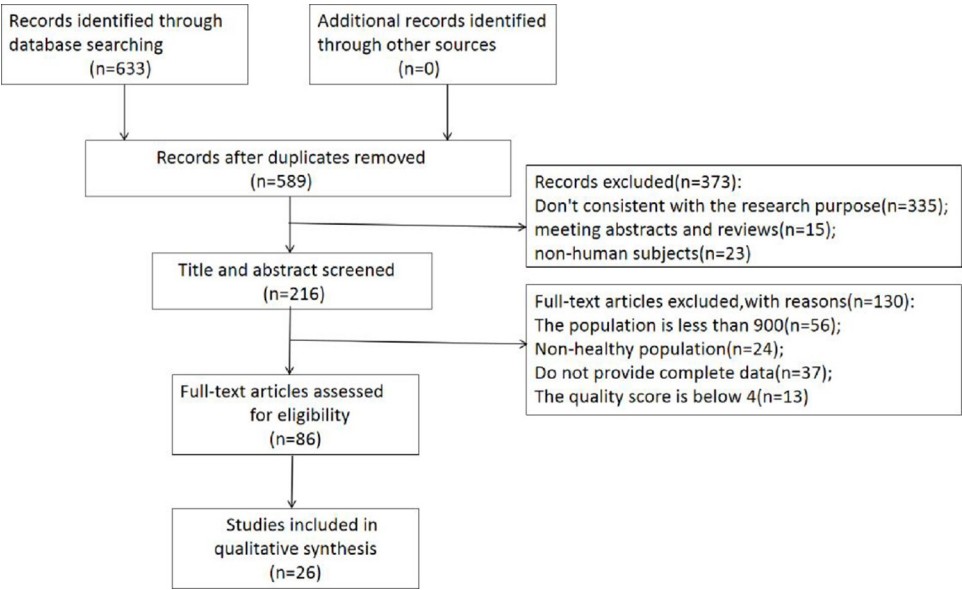

**Fig 1. The process of selecting studies on vitamin D levels in Chinese pregnant women.**

in all parts of China was found to be 84% (95%CI: 81%~88%). This result is presented in forest plot shown in **Fig 2**.

### 3.3 Regional distribution

Out of the 26 studies included in meta-analysis, 17 were conducted in East China, 1 in Northeast China, 3 in North China, 3 in South Central China, 1 in Northwest China, and 1 in Southern China. Heterogeneity test demonstrated statistical significance (P<0.1) and moderate heterogeneity ($I^2$>50%), indicating the use of a random effects model for meta-analysis. Its outcomes indicate that the incidence of insufficient or deficient vitamin D in the process of pregnancy varied among varied regions of China. Furthermore, the distinction in vitamin D deficiency or insufficiency incidence during pregnancy among the six regions was statistically significant ($\chi2$ = 844.80, P<0.001). These findings are presented in the forest plot shown in **Fig 3**.

### 3.4 Trends in recent years

Among the literature included in meta-analysis, 9 articles were before 2016, 4 articles in 2018, 5 articles in 2019–2020, 5 articles in 2021 and 3 articles in 2022. Heterogeneity test results demonstrated P<0.1,$I^2$>50%, so meta-analysis was conducted via random effects model. Meta-analysis results demonstrated the distinction in vitamin D deficiency or insufficiency prevalence in pregnant women in recent years was not statistically significant ($\chi2$ = 4.80,P = 0.31), with forest plot referring to **Fig 4**.

### 3.5 Seasonal distribution

24 studies analyzed seasonal variations in vitamin D, 12 from summer and fall and 12 from winter and spring. The test for heterogeneity showed a P<0.1,$I^2$>50%, so a random-effects model was used for meta-analysis. Meta-analysis demonstrated vitamin D insufficiency or deficiency prevalence in pregnancy was higher in winter and spring compared to that in fall

**Table 1. The basic characteristics of the included literature in the meta-analysis.**

| References | AHRQscores | Region | Province | Sample size | VD insufficient or deficient | Incidence(%) |
|---|---|---|---|---|---|---|
| Sun et al.2018 [19] | 7 | East China | Zhejiang | 1352 | 1027 | 75.96 |
| Shen et al.2019 [20] | 7 | East China | Jiangsu | 1655 | 1374 | 83.02 |
| Qian et al.2021 [18] | 6 | East China | Zhejiang | 3694 | 2833 | 76.69 |
| Zhang et al.2018 [22] | 8 | North China | Neimenggu | 1320 | 898 | 68.03 |
| Jiao et al.2019 [24] | 7 | North China | Beijing | 17351 | 10775 | 62.10 |
| Xu et al.2021 [15] | 7 | Northeast China | Liaoning | 7215 | 5722 | 79.31 |
| Chen et al.2013 [2] | 6 | East China | Zhejiang | 1899 | 1698 | 89.42 |
| Pan et al.2018 [26] | 6 | East China | Jiangsu | 1255 | 1026 | 81.75 |
| Pan et al.2022 [27] | 7 | East China | Zhejiang | 2614 | 2158 | 82.56 |
| Zhong et al.2015 [28] | 7 | East China | Jiangsu | 2722 | 1662 | 61.06 |
| Xu et al.2015 [29] | 8 | East China | Zhejiang | 4666 | 4454 | 95.46 |
| Wang et al.2014 [30] | 7 | South Central China | Guangxi | 7528 | 4197 | 55.75 |
| Lin et al.2016 [31] | 7 | South Central China | Guangdong | 1154 | 1091 | 94.54 |
| Guo et al.2019 [32] | 7 | East China | Shandong | 11193 | 10887 | 97.27 |
| Hu et al.2016 [33] | 6 | East China | Anhui | 3602 | 3598 | 99.89 |
| Ren et al.2020 [34] | 7 | Southwest China | Sichuan | 3346 | 2901 | 86.70 |
| Pei et al.2016 [35] | 6 | East China | Jiangsu | 1499 | 1482 | 98.87 |
| Chen et al.2016 [36] | 7 | East China | Shanghai | 13440 | 13058 | 97.16 |
| Meng et al.2018 [37] | 7 | Northwest China | Shanxi | 10365 | 9758 | 94.14 |
| Gong et al.2019 [38] | 6 | East China | Shanghai | 20657 | 19285 | 93.36 |
| Wu et al.2022 [39] | 6 | North China | Tianjin | 1056 | 1022 | 96.78 |
| Wang et al.2021 [40] | 8 | East China | Shandong | 1643 | 1236 | 75.23 |
| Xu2 et al.2015 [29] | 6 | East China | Jiangsu | 1550 | 1142 | 73.68 |
| Li et al.2021 [42] | 7 | South Central China | Hubei | 2011 | 1516 | 75.39 |
| Chen et al.2022 [43] | 7 | East China | Anhui | 3080 | 3038 | 98.64 |
| Yang et al.2021 [21] | 7 | East China | Shanghai | 953 | 930 | 97.59 |

AHRQ = Agency for Healthcare Research and Quality,VD = Vitamin D.

and summer. The distinction in vitamin D prevalence in disparate seasons was statistically significant ($\chi$2 = 14.77,P<0.001), with forest plot referring to **Fig 5**.

## 3.6 Pregnancy distribution

Among included studies assessing changes in vitamin D levels during pregnancy, 8 studies were from the first trimester, 9 studies from the second trimester, and 8 studies from the third trimester. The test for heterogeneity showed a P<0.1,$I^2$>50%, so meta-analysis was conducted via a random-effects model. Meta-analysis showed that vitamin D insufficiency or deficiency prevalence decreased progressively with pregnancy progression. The difference in vitamin D levels among different pregnant women at different gestational stages was statistically significant ($\chi$2 = 6.18,P = 0.05), with forest plot referring to **Fig 6**.

## 3.7 Sensitivity analysis results

Following excluding small-sample study of Chun et al. [16], overall vitamin D insufficiency or deficiency prevalence among different pregnant women in China was (84%, 95% CI: 0.80–0.87), and the difference was not statistically significant from that before exclusion (84%, 95%

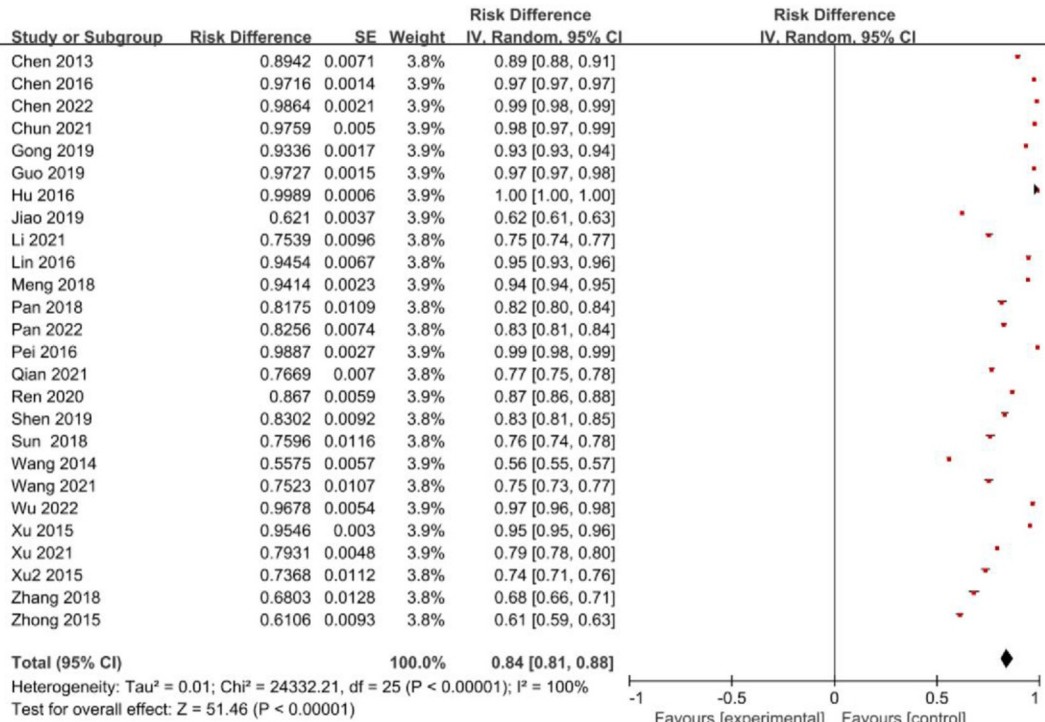

**Fig 2. Forest plot of the prevalence of insufficient or deficient vitamin D among pregnant women in China.**

CI: 0.81–0.88). Therefore, meta-analysis results here were stable, with distinction in sample size not serving as potential influencing factor.

## 3.8 Publication bias

A funnel plot was constructed for the included studies using maternal vitamin D insufficiency or deficiency rates as the indicator. The observed distribution of points in the funnel plot showed a roughly symmetrical clustering at the upper end, indicating the absence of significant publication bias. The funnel plot is shown in **Fig 7**.

## 4. Discussion

Our research indicates that low vitamin D levels among pregnant women is one public health issue in mainland China. Current global estimates suggest that vitamin D deficiency has influenced one billion people in the world, with women of childbearing age and pregnant women being particularly susceptible [44]. Vitamin D insufficiency or deficiency prevalence in the process of pregnancy varies by region: 27.0%~91.0% in U.S.A. 45.0%~100.0% in Asia, 39.0% ~65.0% in Canada, 19.0%~96.0% in Europe, 25.0%~87.0% in Australia and New Zealand [45]. As the first systematic analysis of vitamin D insufficiency or deficiency in the process of pregnancy in mainland China, our data showed a prevalence of 84% (95%CI: 81%~88%), similar to other places in the world. However, maternal vitamin D status can be influenced by multiple factors, like the mother's ethnicity, education level, body mass index (BMI) before pregnancy, sun exposure, pregnancy season, usage of vitamin D supplements as well as local customs regarding clothing habits [46,47].It's important to note that our current research did not

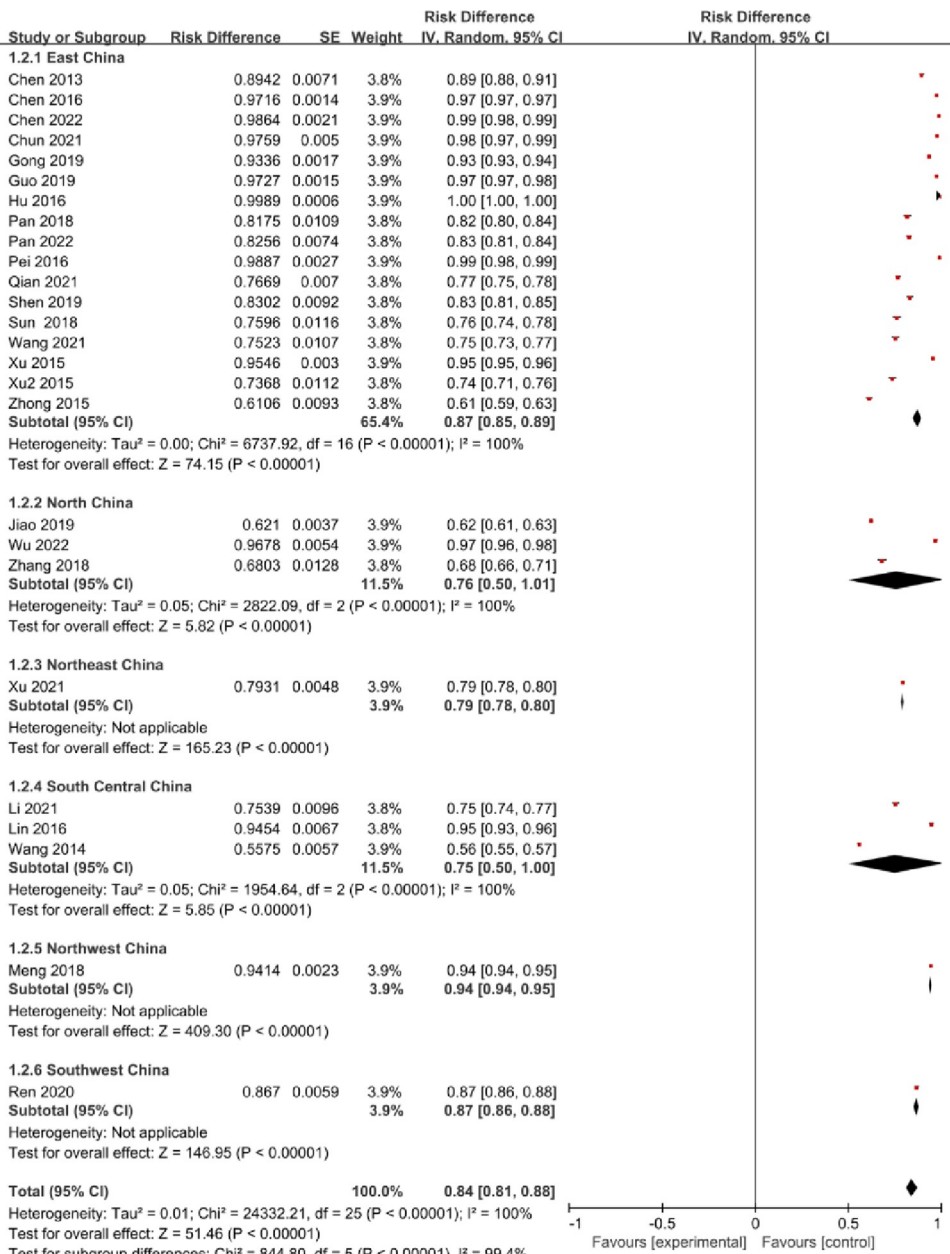

**Fig 3. Forest plot for the regional distribution of the prevalence of insufficient or deficient vitamin D among pregnant women in China.**

collect specific data related to these factors, and therefore, we are unable to provide empirical evidence to support their contribution to the significant differences observed in our study.

Genetic factors also influence maternal vitamin D during pregnancy. Sampathkumar et al. reported a genome-wide association analysis (GWAS) of vitamin D in maternal and fetal blood circulation. The GWAS analysis revealed that a missense variant, rs4588, and its associated haplotype in the population-specific component gene encoding Vitamin D binding protein (VDBP), serve as risk factors for prenatal vitamin D deficiency and low umbilical cord blood vitamin D levels. Furthermore, a novel association was discovered between a

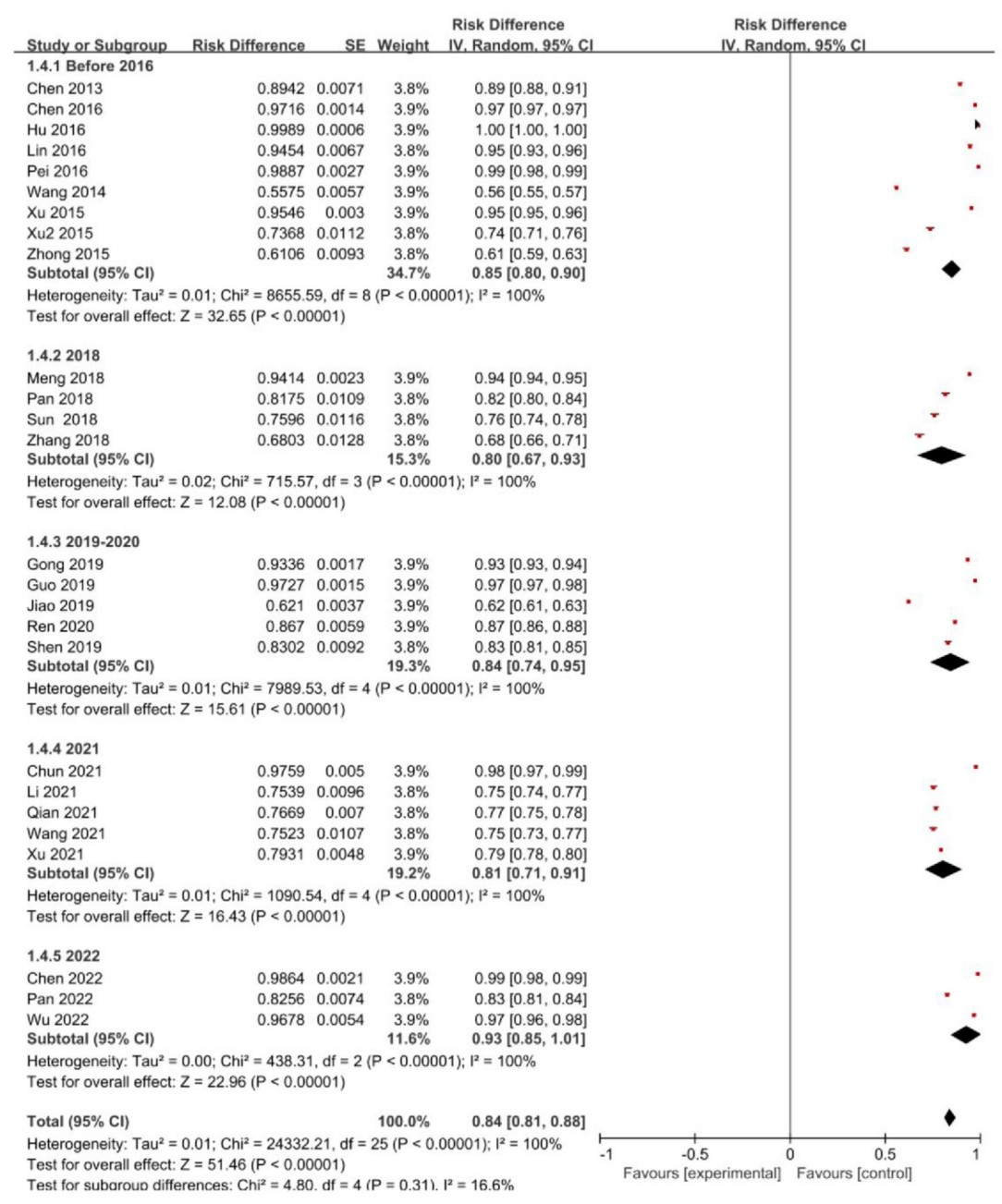

**Fig 4. Forest plot for the trends in recent years of the prevalence of insufficient or deficient vitamin D among pregnant women in China.**

downstream SNP of the CYP2J2 gene (rs10789082), involved in vitamin D 25-hydroxylation, and maternal vitamin D, but this association was not observed in their offspring [48].

Low maternal vitamin D levels during pregnancy are associated with various adverse obstetric outcomes, such as maternal osteomalacia, gestational diabetes, preeclampsia, and cesarean section [49]. Additionally, prenatal vitamin D deficiency is linked to fetal intrauterine growth restriction and a range of adverse fetal and neonatal health outcomes, including

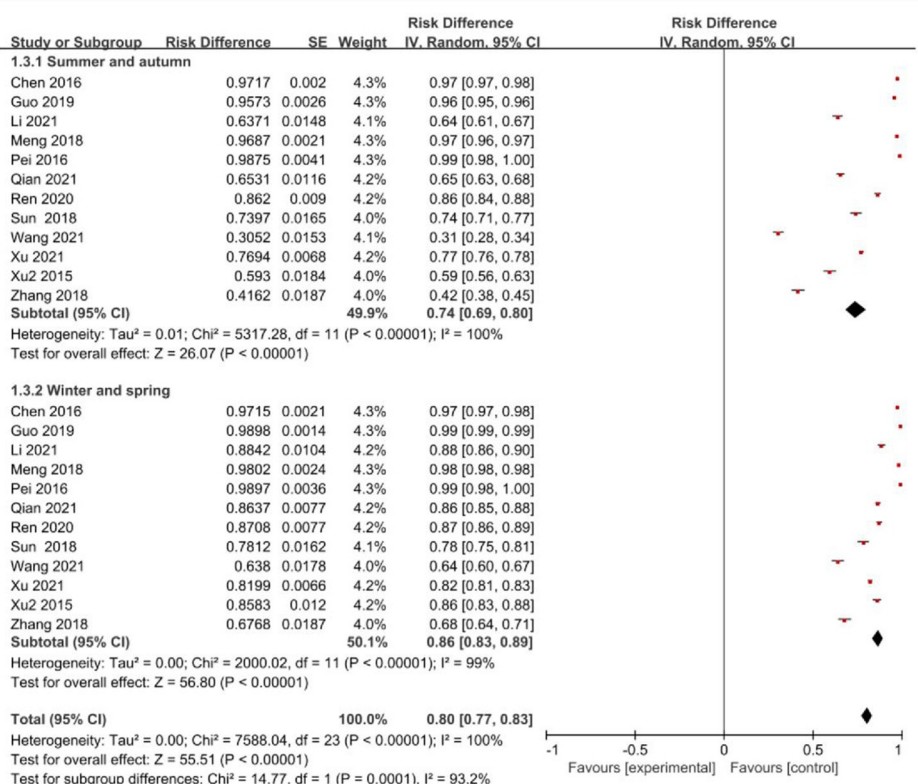

**Fig 5. Forest plot for the seasonal distribution of the prevalence of insufficient or deficient vitamin D among pregnant women in China.**

preterm birth, miscarriage, low birth weight, neonatal hypocalcemia, and an increased risk of childhood obesity [50].

Our subgroup analysis revealed significant differences in the prevalence of vitamin D deficiency among pregnant women in different regions of China. This disparity can be attributed to the vast expanse of China and the substantial regional variations in dietary habits, lifestyles, and economic status [51]. The identification of these regional disparities serves as a reminder for us to consider the influence of geographical factors when formulating policies for vitamin D supplementation in pregnant women. Targeted intervention measures should be implemented to address the vitamin D deficiency among pregnant women in different regions.

In recent years, major economic and social changes in China, especially changes in fertility policies, have led to increased attention to the health of pregnant women. Unfortunately, however, we found no improvement in vitamin D levels in pregnant women. One possible explanation is the prevalence of a cultural preference for fair skin in Asian populations, which may lead to insufficient sun exposure as well as outdoor activity among pregnant women [52].

According to a study in Beijing, vitamin D deficiency prevalence (25-(OH)D < 50 nmol/L) among pregnant women in winter was high at 44.8% [53]. Another research conducted in Nanjing showed that vitamin D deficiency prevalence (25-(OH)D ≤ 20 ng/mL) in pregnant women is 94.7% in summer whereas 96.1% in winter [54]. A Hangzhou survey demonstrated the lowest levels of 25-(OH)D were observed in spring, while the highest levels were found in autumn. In spring, 98.6% of pregnant women realized the level of 25-(OH)D < 50 nmol/L, compared to 64.0% in autumn [55]. This study found a significantly higher vitamin D

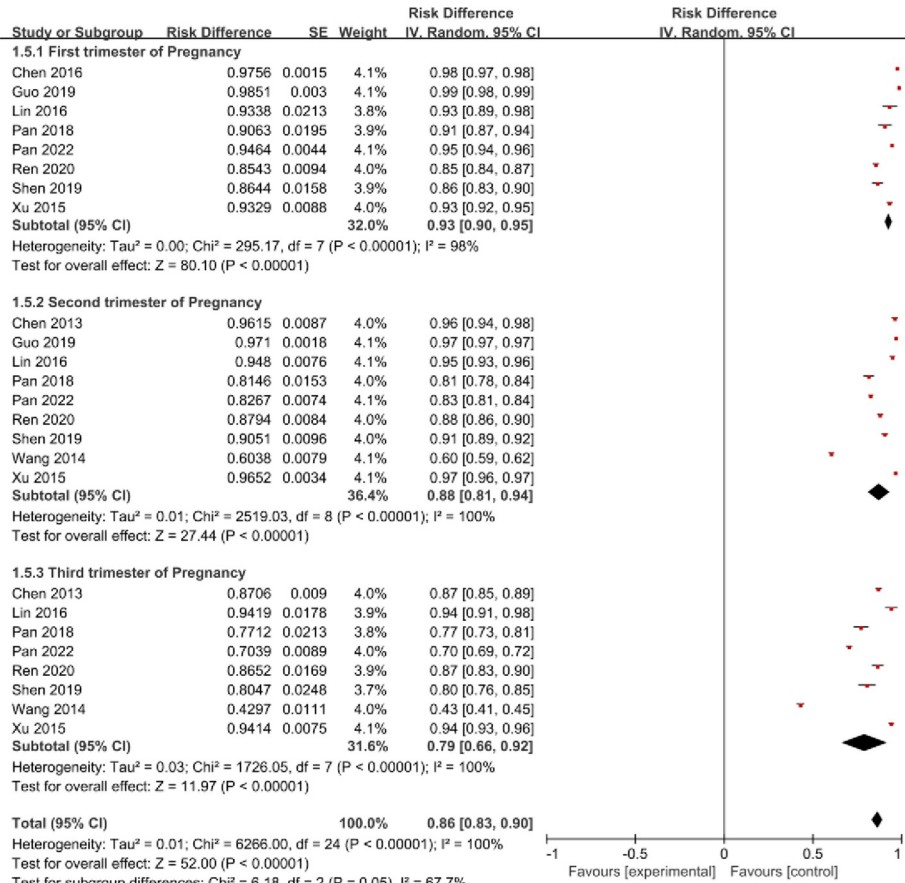

**Fig 6. Forest plot for the pregnancy distribution of the prevalence of insufficient or deficient vitamin D among pregnant women in China.**

insufficiency or deficiency prevalence among pregnant women in winter and spring than summer and fall, suggesting differences in results across studies.

A study in Guiyang City showed that vitamin D insufficiency prevalence was 61.9% and 38.1% in the middle and second trimesters of pregnancy, with 25-(OH)D concentrations of 13.49 ± 5.83 ng/mL and 16.39 ± 7.71 ng/mL, respectively [56]. A similar conclusion was reached in a study in Zhoushan, where vitamin D deficiency prevalence gradually reduced from early pregnancy (65.26%) to mid-gestation (33.56%) and late pregnancy (32%) [57]. According to the results of our study, we observed a gradual reduction in vitamin D deficiency or insufficiency prevalence at different stages of pregnancy. This trend may be related to several factors. First, pregnant women often increase their outdoor activities during pregnancy, which increases their exposure to sunlight. Second, pregnant women are usually more aware of their nutritional status and may consume vitamin D-rich foods, like fatty fish (mackerel, salmon, etc.), eggs and fortified dairy products, for ensuring adequate vitamin D intake.

This study outlines the general vitamin D status among pregnant women in mainland China. However, when interpreting our research findings, several limitations should be considered. Firstly, our study exhibits certain sources of heterogeneity, including variations in sampling methods, specimen types, sampling intervals, and detection techniques, as well as disparities in economic development levels between rural and urban areas. Secondly, in

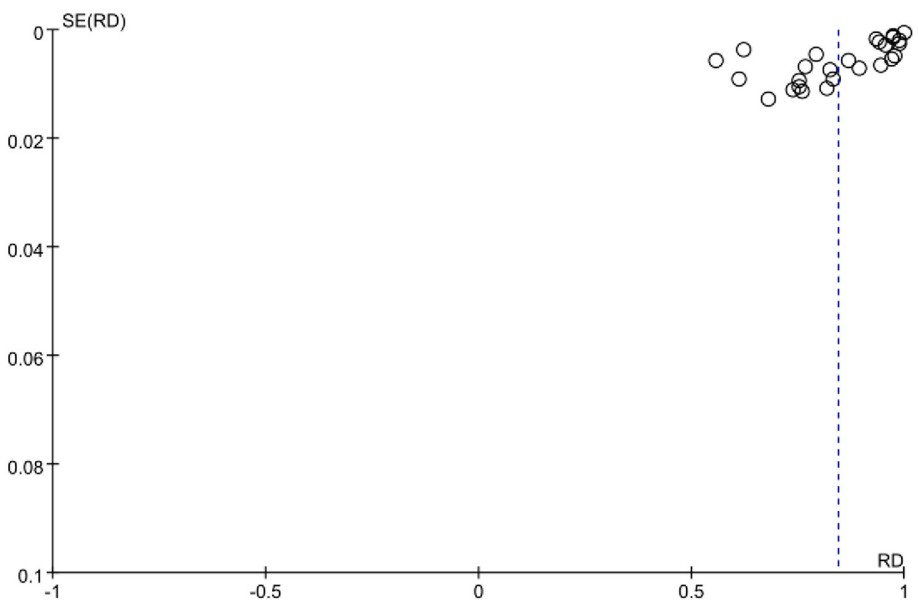

**Fig 7. Funnel plot analysis of maternal vitamin D insufficiency or deficiency rates.**

conducting regional subgroup analyses, the limited number of eligible literature resulted in some regions having only one included study and no further analysis in the geographical analysis, thereby increasing the uncertainty of the results and the level of uncertainty in the conclusions. Thirdly, time impact on vitamin D results was not observed, probably because our study only covered limited time span.Finally,due to limitations in the available data and the scope of our study, our study did not specifically examine the significance of vitamin D deficiency or insufficiency during different periods of pregnancy among different regions and seasons. Therefore, it is necessary to understand and interpret our research findings in light of these limitations.

In summary, our findings highlight the prominent issue of vitamin D insufficiency or deficiency in pregnant women from mainland China and explore associated influencing factors. This may contribute to greater attention from public health experts and officials, encouraging the implementation of targeted investigation and control strategies for maternal vitamin D status. Given unique characteristics of such a population and the potential risks associated with deficiencies, we advocate for the provision of comprehensive guidelines on vitamin D supplementation specifically tailored for the maternal.

## Supporting information

**S1 Checklist. PRISMA 2020 checklist.**
(DOCX)

**S1 File.**
(CSV)

## Acknowledgments

We would like to sincerely thank Master Wang and Prof. Li for their invaluable guidance and support throughout this study. We also express our gratitude to Pengyun Ji and Wenyan Qin

for their valuable contributions to the manuscript revision and data verification. Furthermore, we are grateful to the experts and scholars who generously provided research data and information on vitamin D deficiency in Chinese pregnant women, which laid a strong foundation for this meta-analysis. Additionally, we appreciate the hard work and contributions of our research assistants and colleagues in data collection and organization, which greatly enhanced the comprehensiveness and reliability of this study.

## Author Contributions

**Conceptualization:** Bo Chen, Zisheng Li.

**Data curation:** Bo Chen, Pengyun Ji, Qing Wang, Wenyan Qin.

**Formal analysis:** Qing Wang.

**Resources:** Zisheng Li.

**Software:** Qing Wang.

**Writing – review & editing:** Bo Chen, Pengyun Ji, Wenyan Qin, Zisheng Li.

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
