## [Decision Letter · Decision Letter 0]

26 Oct 2023

PONE-D-23-31837Vitamin D Levels and Its Influencing Factors in Pregnant Women in Mainland China：a systematic review and meta-analysisPLOS ONE

Dear Dr. Li,

Thank you for submitting your manuscript to PLOS ONE. After careful consideration, we feel that it has merit but does not fully meet PLOS ONE’s publication criteria as it currently stands. Therefore, we invite you to submit a revised version of the manuscript that addresses the points raised during the review process.

We look forward to receiving your revised manuscript.

Kind regards,

Linglin Xie

Academic Editor

PLOS ONE

Journal Requirements:

Reviewers' comments:

Reviewer's Responses to Questions

**Comments to the Author**

1. Is the manuscript technically sound, and do the data support the conclusions?

Reviewer #1: Yes

Reviewer #2: Yes

Reviewer #3: Yes

2. Has the statistical analysis been performed appropriately and rigorously? 

Reviewer #1: N/A

Reviewer #2: Yes

Reviewer #3: Yes

3. Have the authors made all data underlying the findings in their manuscript fully available?

Reviewer #1: Yes

Reviewer #2: Yes

Reviewer #3: Yes

4. Is the manuscript presented in an intelligible fashion and written in standard English?

Reviewer #1: Yes

Reviewer #2: Yes

Reviewer #3: Yes

5. Review Comments to the Author

Reviewer #1: In this study, the authors performed a systematic review and meta-analysis to comprehensively understand the prevalence of Vitamin D deficiency among pregnant women in mainland China. This study will provide some guidance for vitamin D supplementation of pregnant women. I have several questions as following:

1. The author also mentioned that the prevalence of vitamin D deficiency or insufficiency decreased during pregnancy progression. The author showed that the distinction in vitamin D deficiency or insufficiency during pregnancy among different regions and seasons is significant, as shown in Fig. 3 and Fig. 5, respectively. Could the authors clarify which pregnancy period’s vitamin D deficiency or insufficiency is significantly different among different regions and seasons?

2. How did the author define vitamin D deficiency or insufficiency? Since the requirement of vitamin D changes during pregnancy progression, did the author use different standards of vitamin D levels in different pregnancy periods?

3. The author indicated that the difference in vitamin D levels among different pregnant women at different gestational stages was significant (P<0.05). However, Fig. 6 indicates that P=0.05. Please double check this inconsistency.

4. As the authors mentioned in discussion, maternal vitamin D status can be influenced by several factors, such as ethnicity, education level, BMI, sun exposure, pregnancy season, and usage of vitamin D supplements. Do the author have data to support the potential contribution of these factors to the significant differences observed in this study? The author should discuss more details in this regard.

5. Genetic factor may also contribute to the differences observed in this study. The author should discuss it.

Reviewer #2: This article is titled “Vitamin D Levels and Its Influencing Factors in Pregnant Women in Mainland China：a systematic review and meta-analysis” analyzed the VD deficiency during pregnancy in multiple angles, which is very interesting and meaningful, filling the knowledge gap in this field regarding the Chinese population in Mainland China. However, I still have some concerns that I hope the authors can address for publication:

1. The chemical properties of vitamin D should be introduced.

2. What is the normal/healthy range of VD during pregnancy? What is the difference between “insufficiency” and “deficiency”?

3. Due to the VD deficiency during pregnancy, were there any consequences or symptoms shown on the offspring during pregnancy or after delivery? Please give an adequate discussion.

Reviewer #3: Summary

In this manuscript, Chen et al. performed a thorough meta-analysis and sensitivity analysis to examine the prevalence and potential factors of maternal vitamin D deficiency in mainland China. With a robust criterion for selecting high-quality literature, the manuscript provides a relatively accurate reflection of the statistics over the last decade (2013–2023). The regional variation in vitamin D deficiency highlights some facts about dietary dependence. It is also worth noting that, due to the limitations of the data, resources, and design of the study, some important features or observations were not clearly demonstrated. Overall, the manuscript is well-written. However, there are some flaws in the results that need to be addressed. Please refer to the comments outlined below. Please also proofread the manuscript and correct all noticeable grammatical errors.

Major Comments

• There is a lack of references being cited in the introduction section. As an illustration,

1. Line 48-52, the first paragraph: “Vitamin D is one of essential nutrients critical to metabolic processes in human body…. osteochondrosis.” Please add references.

2. Line 61, “The concentration of 25-(OH)D remained relatively stable in the whole process of pregnancy.”

3. Line 68-71, “Vitamin D regulation of the intrauterine… pregnancy complications.”

• For the Chi squared tests used for heterogeneity analysis, what would the results be if the authors choose a more common significance level alpha = 0.05?

• Based on the literature quality evaluation system, a score between 8 to 11 denotes high quality literature. According to the authors, the study only included the classification of high-quality literature. However, based on Table 1 AHRQ scores, 23 out of the 26 selected studies have a score that falls into the medium quality literature. I suggest the authors make a clearer clarification in section 2.3. For the meta-analysis, 88% of the studies were non-high-quality.

• As the authors mentioned, one of the limitations of the analysis in the regional distribution comparison is the unbalanced number of studies conducted in different regions. They do differ quite a lot so the heterogeneity test might not be too convincing. Is there a way that the authors could subdivide the studies based on a better criterion? For instance, let the categorization be based on cities/provinces instead of national regions. This might improve the results in the heterogeneity test.

• There is some inconsistency regarding the selection criteria. Based on figure 1, studies with population/sample size smaller than 1000 were excluded. However, there is one study in Table 1 that only has 953 population: Chun et al. 2021.

• Although the authors concluded that there are no temporal effects on the vitamin D deficiency, the meta-analysis did not consider regional differences. I suggest the author perform the analyses within each regional subgroup.

• It seems less clear to me how to justify the selection criterion in the sensitivity analysis. The authors mentioned the small-sample study of Wu et al. [36] was excluded on Line 234. However, that study does not have the lowest sample size. It also has a higher incidence (%) than those lower ranked ones. Also reference 36 is not matching the one mentioned above:

[36] Xu YH, Qian YB, Zeng PP, et al. Analysis of the nutritional status of serum 25-hydroxyvitamin D in pregnant women in different seasons in Wuxi region. Hainan Medical Journal 2015;26:68-69.

Please make sure all cited references are in the correct order. Is it referring to [34]?

• If there is only one study conducted in Northwest China, I find it less convincing or sound to conclude that “the highest incidence was found in Northwest China”. This is also one of the major limitations in the study. The same applies to “lowest incidence was observed in South Central China” with only one study included.

• I appreciate the authors mentioning the pivotal factors: “However, maternal vitamin D status can be influenced by multiple factors, like the mother's ethnicity, education level, body mass index (BMI) before pregnancy, sun exposure, pregnancy season, usage of vitamin D supplements as well as local customs regarding clothing habits.” I hope the authors could elaborate more on how these factors could possibly be associated with the regional differences as well as temporal effects on vitamin D deficiency. If necessary, some augmented analyses could also be included.

Minor comments

• Typo Line 130, “ ” Yes”.

• Line 66, improper use of comma. Please correct this run-on sentence. Consider rewriting or breaking it up.

• Grammatical error, Line 99-100. “The ultimate goal…”

• Line 103, missing period.

• Line 163, sentence is misleading. “The selection process refers to Fig 1.” Improper use of “refer to”. Please rephrase.

• Line 153, formatting issue after “random-effects model”.

6. PLOS authors have the option to publish the peer review history of their article (what does this mean?). If published, this will include your full peer review and any attached files.

Reviewer #1: No

Reviewer #2: No

Reviewer #3: No

---

## [Author Response · Author response to Decision Letter 0]

17 Nov 2023

Dear Editor and Reviewers:

Thanks for your letter and for the reviewers’ comments concerning our manuscript entitled “Vitamin D Levels and Its Influencing Factors in Pregnant Women in Mainland China：a systematic review and meta-analysis” (ID: PONE-D-23-31837). Those comments are all valuable and very helpful for revising and improving our paper, as well as the important guiding significance to our researches. We have studied comments carefully and have made correction which we hope meet with approval. Revised portion are marked in red in the paper. The main corrections in the paper and the responds to the reviewer’s comments are as following:

The academic editor:

Journal Requirements:

Response: Thank you very much for your reminder. We have checked it again, and ensured that the manuscript meet PLOS ONE’s style requirements, including those for file naming.

Response: Thank you very much for your reminder. The ORCID iD for the corresponding author (Zisheng Li,) is0009-0005-7807-0046 , and it is validated in Editorial Manager.

Response: Thank you very much for your reminder. The captions for the Supporting Information file was added in the revised manuscript.

“Supporting Information: PRISMA 2020 checklist.” 

Reviewers' comments:

Reviewer's Responses to Questions

Comments to the Author

1. Is the manuscript technically sound, and do the data support the conclusions?

Reviewer #1: Yes

Reviewer #2: Yes

Reviewer #3: Yes

2. Has the statistical analysis been performed appropriately and rigorously?

Reviewer #1: N/A

Reviewer #2: Yes

Reviewer #3: Yes

3. Have the authors made all data underlying the findings in their manuscript fully available?

Reviewer #1: Yes

Reviewer #2: Yes

Reviewer #3: Yes

4. Is the manuscript presented in an intelligible fashion and written in standard English?

Reviewer #1: Yes

Reviewer #2: Yes

Reviewer #3: Yes

5. Review Comments to the Author

Reviewer #1: In this study, the authors performed a systematic review and meta-analysis to comprehensively understand the prevalence of Vitamin D deficiency among pregnant women in mainland China. This study will provide some guidance for vitamin D supplementation of pregnant women. I have several questions as following:

1. The author also mentioned that the prevalence of vitamin D deficiency or insufficiency decreased during pregnancy progression. The author showed that the distinction in vitamin D deficiency or insufficiency during pregnancy among different regions and seasons is significant, as shown in Fig. 3 and Fig. 5, respectively. Could the authors clarify which pregnancy period’s vitamin D deficiency or insufficiency is significantly different among different regions and seasons?

Response: Thank you for your comment and question regarding the distinction in vitamin D deficiency or insufficiency during pregnancy among different regions and seasons. After carefully re-evaluating our findings and figures, we would like to clarify that our study did not specifically examine the significance of vitamin D deficiency or insufficiency during different periods of pregnancy among different regions and seasons. Our analysis focused on investigating the overall prevalence of vitamin D deficiency or insufficiency in pregnant women in mainland China. We acknowledge that there may be variations in vitamin D levels during different stages of pregnancy and among different regions and seasons. However, due to limitations in the available data and the scope of our study, we were unable to conduct a detailed analysis of these specific factors.We have addressed this issue in the limitations section of our paper. Thank you for bringing this to our attention, and we appreciate your understanding. 

In the eighth part of the discussion: “Finally,due to limitations in the available data and the scope of our study, our study did not specifically examine the significance of vitamin D deficiency or insufficiency during different periods of pregnancy among different regions and seasons.”

2. How did the author define vitamin D deficiency or insufficiency? Since the requirement of vitamin D changes during pregnancy progression, did the author use different standards of vitamin D levels in different pregnancy periods?

Response: Thanks very much for reminder. The define vitamin D deficiency or insufficiency was added in the revised manuscript. In our study, we have defined the criteria for vitamin D deficiency or insufficiency and classified the pregnant women participating in the study accordingly. We based our definitions on the "Guidelines for the Application of Vitamin D in Adults' Bone Health" developed by the Vitamin D Discipline Group of the Osteoporosis Committee of the Chinese Geriatrics Society in 2014. According to these guidelines, a serum 25(OH)D level below 30 nmol/L is considered as vitamin D deficiency, and a level between 30-50 nmol/L is considered as vitamin D insufficiency,and vitamin D levels above 50 nmol/L are considered sufficient.

In our study, we used a standardized criterion to define vitamin D deficiency or insufficiency across all stages of pregnancy, without considering different criteria based on the progression of pregnancy. This approach was chosen because our main objective was to investigate the overall vitamin D status of pregnant women, rather than analyzing the variation in vitamin D requirements during specific stages of pregnancy.

However, we acknowledge that the vitamin D requirements may indeed vary during different stages of pregnancy. In future research, it would be valuable to consider the progression of pregnancy and relevant guidelines in order to assess vitamin D deficiency or insufficiency using different criteria for each stage of pregnancy.

In the second part of the materials and methods: “We based our definitions on the "Guidelines for the Application of Vitamin D in Adults' Bone Health" developed by the Vitamin D Discipline Group of the Osteoporosis Committee of the Chinese Geriatrics Society in 2014. According to these guidelines, a serum 25(OH)D level below 30 nmol/L is considered as vitamin D deficiency, and a level between 30-50 nmol/L is considered as vitamin D insufficiency,and vitamin D levels above 50 nmol/L are considered sufficient.”

3. The author indicated that the difference in vitamin D levels among different pregnant women at different gestational stages was significant (P<0.05). However, Fig. 6 indicates that P=0.05. Please double check this inconsistency.

Response: Thank you very much for your care, and we apologize for our mistake. We have checked and revised the sentence in the revised manuscript.

In the sixth part of the result: “The difference in vitamin D levels among different pregnant women at different gestational stages was statistically significant (χ2=6.18,P=0.05), with forest plot referring to Fig 6.”

4. As the authors mentioned in discussion, maternal vitamin D status can be influenced by several factors, such as ethnicity, education level, BMI, sun exposure, pregnancy season, and usage of vitamin D supplements. Do the author have data to support the potential contribution of these factors to the significant differences observed in this study? The author should discuss more details in this regard.

Response: Thanks for your reminder. We appreciate your suggestion to delve further into these aspects in our study. However, it's important to note that our current research did not collect specific data related to these factors, and therefore, we are unable to provide empirical evidence to support their contribution to the significant differences observed in our study.

We agree that understanding the interplay of these variables could provide valuable insights into the variations in maternal vitamin D status. In future research, we plan to investigate these factors more comprehensively to elucidate their roles in influencing vitamin D levels during pregnancy. Your comment has highlighted an important avenue for further investigation, and we appreciate your valuable input.We have discussed this aspect in the revised manuscript.

In the first part of the discussion:“It's important to note that our current research did not collect specific data related to these factors, and therefore, we are unable to provide empirical evidence to support their contribution to the significant differences observed in our study.”

5. Genetic factor may also contribute to the differences observed in this study. The author should discuss it.

Response: Thanks for your good suggestion. We have added a paragraph for clarification in the second part of discussion.

In the second part of the discussion: “Genetic factors also influence maternal vitamin D during pregnancy. Sampathkumar et al. reported a genome-wide association analysis (GWAS) of vitamin D in maternal and fetal blood circulation. The GWAS analysis revealed that a missense variant, rs4588, and its associated haplotype in the population-specific component gene encoding Vitamin D binding protein (VDBP), serve as risk factors for prenatal vitamin D deficiency and low umbilical cord blood vitamin D levels. Furthermore, a novel association was discovered between a downstream SNP of the CYP2J2 gene (rs10789082), involved in vitamin D 25-hydroxylation, and maternal vitamin D, but this association was not observed in their offspring.” 

Special thanks to you for your good comments.

Reviewer #2: This article is titled “Vitamin D Levels and Its Influencing Factors in Pregnant Women in Mainland China：a systematic review and meta-analysis” analyzed the VD deficiency during pregnancy in multiple angles, which is very interesting and meaningful, filling the knowledge gap in this field regarding the Chinese population in Mainland China. However, I still have some concerns that I hope the authors can address for publication:

1. The chemical properties of vitamin D should be introduced.

Response: Thanks very much for your good suggestion. We have added some sentences to introduce the chemical properties of vitamin D in the first part of the introduction.

In the first part of the introduction: “The active form of vitamin D in the body is known as 1,25-dihydroxyvitamin D. In the human body, vitamin D3 and D2 bind to vitamin D binding protein in the blood plasma and are transported to the liver, where they undergo 25-hydroxylation to become 25-hydroxyvitamin D [25(OH)D].” 

2. What is the normal/healthy range of VD during pregnancy? What is the difference between “insufficiency” and “deficiency”?

Response: Thanks very much for reminder. Currently, there are no specific standards for the normal/healthy range of vitamin D in pregnant women; therefore, our criteria are based on those established for healthy adults.The define vitamin D normal/health range during pregnancy, deficiency or insufficiency was added in the revised manuscript.

In the second part of the materials and methods: “We based our definitions on the "Guidelines for the Application of Vitamin D in Adults' Bone Health" developed by the Vitamin D Discipline Group of the Osteoporosis Committee of the Chinese Geriatrics Society in 2014. According to these guidelines, a serum 25(OH)D level below 30 nmol/L is considered as vitamin D deficiency, and a level between 30-50 nmol/L is considered as vitamin D insufficiency,and vitamin D levels above 50 nmol/L are considered sufficient.”

3. Due to the VD deficiency during pregnancy, were there any consequences or symptoms shown on the offspring during pregnancy or after delivery? Please give an adequate discussion.

Response: Thanks for your good suggestion. We have added a paragraph for clarification in the third part of discussion.

In the third part of the discussion: “Low maternal vitamin D levels during pregnancy are associated with various adverse obstetric outcomes, such as maternal osteomalacia, gestational diabetes, preeclampsia, and cesarean section. Additionally, prenatal vitamin D deficiency is linked to fetal intrauterine growth restriction and a range of adverse fetal and neonatal health outcomes, including preterm birth, miscarriage, low birth weight, neonatal hypocalcemia, and an increased risk of childhood obesity.” 

Special thanks to you for your good comments.

Reviewer #3: Summary

In this manuscript, Chen et al. performed a thorough meta-analysis and sensitivity analysis to examine the prevalence and potential factors of maternal vitamin D deficiency in mainland China. With a robust criterion for selecting high-quality literature, the manuscript provides a relatively accurate reflection of the statistics over the last decade (2013–2023). The regional variation in vitamin D deficiency highlights some facts about dietary dependence. It is also worth noting that, due to the limitations of the data, resources, and design of the study, some important features or observations were not clearly demonstrated. Overall, the manuscript is well-written. However, there are some flaws in the results that need to be addressed. Please refer to the comments outlined below. Please also proofread the manuscript and correct all noticeable grammatical errors.

Major Comments

• There is a lack of references being cited in the introduction section. As an illustration,

1. Line 48-52, the first paragraph: “Vitamin D is one of essential nutrients critical to metabolic processes in human body…. osteochondrosis.” 

---

## [Decision Letter · Decision Letter 1]

10 Jan 2024

Vitamin D Levels and Its Influencing Factors in Pregnant Women in Mainland China：a systematic review and meta-analysis

PONE-D-23-31837R1

Dear Dr. Li,

We’re pleased to inform you that your manuscript has been judged scientifically suitable for publication and will be formally accepted for publication once it meets all outstanding technical requirements.

Kind regards,

Linglin Xie

Academic Editor

PLOS ONE

Additional Editor Comments (optional):

Reviewers' comments:

Reviewer's Responses to Questions

**Comments to the Author**

1. If the authors have adequately addressed your comments raised in a previous round of review and you feel that this manuscript is now acceptable for publication, you may indicate that here to bypass the “Comments to the Author” section, enter your conflict of interest statement in the “Confidential to Editor” section, and submit your "Accept" recommendation.

Reviewer #1: All comments have been addressed

Reviewer #2: All comments have been addressed

Reviewer #3: All comments have been addressed

2. Is the manuscript technically sound, and do the data support the conclusions?

Reviewer #1: Yes

Reviewer #2: Yes

Reviewer #3: Yes

3. Has the statistical analysis been performed appropriately and rigorously? 

Reviewer #1: Yes

Reviewer #2: I Don't Know

Reviewer #3: Yes

4. Have the authors made all data underlying the findings in their manuscript fully available?

Reviewer #1: Yes

Reviewer #2: Yes

Reviewer #3: Yes

5. Is the manuscript presented in an intelligible fashion and written in standard English?

Reviewer #1: Yes

Reviewer #2: Yes

Reviewer #3: Yes

6. Review Comments to the Author

Reviewer #1: (No Response)

Reviewer #2: (No Response)

Reviewer #3: I appreciate the authors' delicate efforts to address all my comments.

The manuscript has been properly revised and I recommend for publication.

However, a few formatting issues should be addressed during the final proof:

1. The newly inserted references should be properly formatted, with added spaces between the brackets and main text.

For instance, Lines 53-76. 274-296.

2. There are some places where missing period(s) are being noticed. (Line 65, after [5-6].)

7. PLOS authors have the option to publish the peer review history of their article (what does this mean?). If published, this will include your full peer review and any attached files.

Reviewer #1: No

Reviewer #2: No

Reviewer #3: No
